# Microtubule assembly governed by tubulin allosteric gain in flexibility and lattice induced fit

**Maxim Igaev\*, Helmut Grubmüller\***

Department of Theoretical and Computational Biophysics, Max Planck Institute for Biophysical Chemistry, Göttingen, Germany

**Abstract** Microtubules (MTs) are key components of the cytoskeleton and play a central role in cell division and development. MT assembly is known to be associated with a structural change in $\alpha\beta$-tubulin dimers from kinked to straight conformations. How GTP binding renders individual dimers polymerization-competent, however, is still unclear. Here, we have characterized the conformational dynamics and energetics of unassembled tubulin using atomistic molecular dynamics and free energy calculations. Contrary to existing allosteric and lattice models, we find that GTP-tubulin favors a broad range of almost isoenergetic curvatures, whereas GDP-tubulin has a much lower bending flexibility. Moreover, irrespective of the bound nucleotide and curvature, two conformational states exist differing in location of the anchor point connecting the monomers that affects tubulin bending, with one state being strongly favored in solution. Our findings suggest a new combined model in which MTs incorporate and stabilize flexible GTP-dimers with a specific anchor point state.

DOI: https://doi.org/10.7554/eLife.34353.001

**\*For correspondence:**
migaev@mpibpc.mpg.de (MI);
hgrubmu@gwdg.de (HG)

**Competing interests:** The authors declare that no competing interests exist.

## Introduction

Microtubules (MTs) are dynamic cytoskeletal filaments formed from $\alpha\beta$-tubulin heterodimers that are abundant in eukaryotic cells and involved in many processes that are essential for cell physiology, for example, cell division and neural growth (*Hyams and Lloyd, 1994*). Unlike other cytoskeletal components in the cell, for example, actin filaments which continuously grow as long as enough G-actin is present, MTs stochastically switch between growing and shrinking phases even under sufficient free tubulin concentrations. This *dynamic instability* is observed both in vitro and in living cells and allows the MT cytoskeleton to be rapidly remodelled in response to internal and external signals (*Mitchison and Kirschner, 1984*; *Gardner et al., 2011*).

Free tubulin requires guanosine triphosphate (GTP) to initiate the formation of new MTs (nucleation) and elongate already existing ones (growth) (*Weisenberg et al., 1968*; *Weisenberg, 1972*). In the latter case, GTP-tubulin stacks head-to-tail at the tips of growing MTs (primarily those capped by $\beta$-tubulin) and forms one-dimensional protofilaments that associate laterally to create a hollow cylinder typically comprising 13–14 protofilaments. Every dimer contains two GTP-binding sites: one nonexchangeable site that is buried in the intradimer contact and one exchangeable site exposed on the outer surface of the $\beta$-subunit. Upon addition of a tubulin dimer to a growing MT end, $\alpha$-tubulin in the incoming dimer completes the GTP-binding pocket of $\beta$-tubulin in the preceding dimer, which enables the hydrolysis of GTP in the pocket, *i.e.* cleavage of GTP into a guanosine diphosphate (GDP) and a $\gamma$-phosphate. MTs grow steadily as long as GTP hydrolysis in the lattice lags behind the arrival of new dimers, which creates the so-called GTP cap at the MT tip (*Mitchison and Kirschner, 1984*). If a MT occasionally grows at a slower rate than GTP hydrolysis, the MT lattice depolymerizes rapidly. This catastrophe is opposed by a rescue mechanism, possibly

due to 'islands' of GTP-dimers embedded in the depolymerizing MT lattice, which serve as nucleation checkpoints (*Dimitrov et al., 2008*). GDP-tubulin released during depolymerization exchanges GDP for GTP and becomes again competent for polymerization and nucleation.

Despite decades of intensive research, the precise mechanism of how the conformational dynamics of tubulin contributes to MT assembly remains elusive (*Brouhard, 2015*). Analysis of electron microscopy images of MTs and structural data have revealed that tubulin dimers are straight when locked in the MT lattice (*Nogales et al., 1998*; *Li et al., 2002*), slightly kinked ($\sim 5°$ per dimer) at the tips of growing MTs (*Chrétien et al., 1995*), and strongly kinked outwards ($\sim 12°$ per dimer) at the tips of shrinking MTs (*Mandelkow et al., 1991*; *Arnal et al., 2000*). X-ray crystallography and cryo-electron microscopy (cryo-EM) structures support this evidence and can be tentatively assigned to each of the above polymerization states (*Nogales et al., 1998*; *Löwe et al., 2001*; *Gigant et al., 2000*; *Wang and Nogales, 2005*). It has hence become clear that tubulin straightening takes place during polymerization, whereas strongly kinked conformations are characteristic of depolymerized GDP-tubulin. Nevertheless, the role of GTP binding in the complex process of MT assembly still remains unclear. One of the central questions being controversially discussed is: does unassembled tubulin exist in different nucleotide-dependent conformational states which may modulate its polymerization kinetics?

The debate of whether tubulin conformation is or is not determined by the nucleotide state of its β-subunit originated from the studies of small tubulin oligomers and rings formed during cold depolymerization of pre-assembled MTs (*Melki et al., 1989*; *Müller-Reichert et al., 1998*). According to the *allosteric model* (*Figure 1*, right branch), there is an equilibrium between two conformations of free tubulin, straight and kinked, under the allosteric control of the nucleotide state. Here, free tubulin dimers bind GTP prior to assembly, which triggers a kinked-to-straight conformational change such that dimer integration into the MT lattice is sterically compatible. This model has been supported by the observation of an intermediate curvature state of tubulin at low temperatures using a nonhydrolyzable GTP analog (*Müller-Reichert et al., 1998*; *Wang and Nogales, 2005*). Alternatively, the *lattice model* (*Figure 1*, left branch) postulates that free tubulin adopts a kinked conformation in solution irrespectively of the nucleotide state, and only upon integration into the MT lattice, tubulin dimers are forced into a straight conformation. Here, the role of GTP binding is not to control the intrinsic tubulin conformation but to strengthen the dimer-dimer bonds in the lattice. This model relies on observations from MT drug binding assays and small-angle X-ray scattering (SAXS) experiments (*Manuel Andreu et al., 1989*; *Buey et al., 2006*; *Rice et al., 2008*). Thus, the main difference between the two models is that the allosteric model treats the kinked-to-straight transition in tubulin as a *cause* and the lattice model as a *consequence* of polymerization. This difference originates from the fact that the unconstrained (MT-free) dynamics of GTP-tubulin are largely unclear. For the lattice or allosteric model to hold, different properties of unassembled dimers are required. Therefore, depending on the allosteric response of GTP-tubulin in solution compared to GDP-tubulin, it should be possible to rule out one or both canonical models, at least in their original form.

Currently, indirect evidence for both models exists. Crystal structures of unassembled tubulin in all relevant nucleotide states, albeit bound to MT-depolymerizing proteins or drugs, have been resolved by several research groups (*Gigant et al., 2005*; *Nawrotek et al., 2011*; *Ayaz et al., 2012*; *Prota et al., 2013*). These structures did not show marked differences between the different nucleotide states, thus supporting the lattice model. The intrinsic conformational dynamics of both GTP- and GDP-tubulin in solution have also been simulated for several tens of nanoseconds using all-atom molecular dynamics (MD) and both were found to be kinked (*Gebremichael et al., 2008*). Longer MD simulations (up to 100 ns) have not revealed GTP-induced tubulin straightening either (*Bennett et al., 2009*; *Grafmüller and Voth, 2011*; *André et al., 2012*; *Grafmüller et al., 2013*). In another computational study, lateral binding of two GDP-tubulin dimers has been shown to shift the conformations of individual dimers toward the straight state; the GTP-state was not analyzed though (*Peng et al., 2014*). The available structural data combined with the simulation evidence therefore indirectly favor the lattice model. However, external factors (*e.g.*, MT-affecting drugs or proteins) are still critical for the crystallization of tubulin, whereas nanosecond time scales of tubulin dynamics so far covered by MD simulations might be too short for large-scale nucleotide-induced effects to occur.

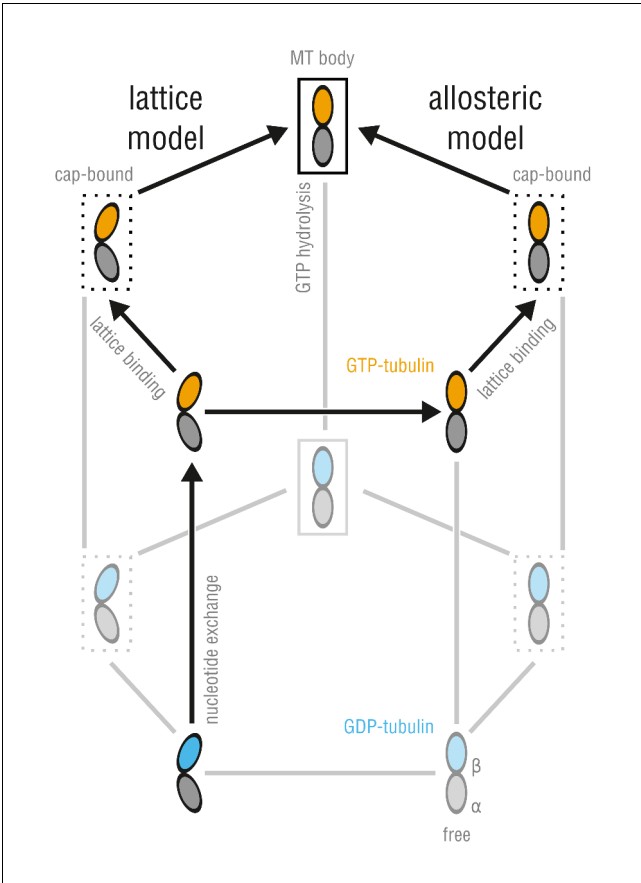

**Figure 1.** Allosteric and lattice models of MT assembly represented by a three-dimensional thermodynamic cycle. GTP- (orange) and GDP-dimers (blue) can be free (no box), MT-cap-bound (dotted box), or integrated into the MT body (solid box). The dimmed states denote energetically unfavored states.
DOI: https://doi.org/10.7554/eLife.34353.002

Recently, the allosteric model has been supported by tubulin polymerization assays and structural experiments which showed that two $\beta$-tubulin mutations (D417H and R262H) related to ocular motility disorder in humans substantially affect MT dynamics (*Ti et al., 2016*). Interestingly, while these mutations were not seen to cause any substantial changes within the MT lattice, they significantly promoted MT polymerization. *Ti et al., 2016* have concluded that the mutations might exploit an allosteric mechanism that reduces the fraction of kinked *vs.* straight tubulin dimers relative to wild-type tubulin. Another $\beta$-tubulin mutation (T238A) known to yield hyperstable MTs in yeast has been recently shown to uncouple the conformational dynamics of tubulin from its GTP-hydrolyzing activity in MTs (*Machin et al., 1995*; *Geyer et al., 2015*). The authors demonstrated that this mutation also reduces the propensity of tubulin dimers bound to the MT cap to be kinked. Although the observations of *Ti et al., 2016* and *Geyer et al., 2015* are indirect and their relation to GTP hydrolysis is not yet clear, they suggest that at least a certain degree of allosteric regulation through GTP binding may still be beneficial for tubulin to enter the MT lattice.

In light of these new findings and the absence of strong evidence for or against either of the current models, the dynamics and flexibility of unassembled tubulin in solution need closer investigation. Here, we address the effect of the nucleotide state on the intrinsic dynamics and energetics of tubulin in the absence of external factors (*e.g.*, MT-affecting drugs or proteins) by all-atom, explicit-solvent conformation and free energy calculations. Currently, high-resolution GTP- and GDP-tubulin structures in various curvature states are available, allowing direct simulations of the corresponding conformational basins without the need for manual nucleotide cleavage or docking. We have performed multiple microsecond long simulations of a tubulin dimer in solution in two different

incarnations, GTP- and GDP-tubulin, and compared them to a set of all tubulin structures currently available in the Protein Data Bank (PDB). Analysis of these simulations has identified two states associated with nucleotide-dependent modes of tubulin bending and separated by a high free energy barrier, and has enabled us to quantify their energetics and relative occupancies under equilibrium conditions. Our results resolve previous seemingly contradictory findings and suggest a new combined mechanism of MT assembly.

## Results

### Free tubulin dynamics depend on the nucleotide state

To study how the nucleotide state affects the dynamics and intrinsic bending of tubulin, we performed multiple microsecond long MD simulations starting from tubulin structures in two different nucleotide states: GTP and GDP (see Materials and methods). For each nucleotide state, two independent $3\mu s$-long simulations were started from a straight and kinked conformation, respectively, yielding a total of $2 \times (2 \times 3\,\mu s) \times 2 = 24\,\mu s$ of tubulin dynamics.

In order to compare the simulated tubulin ensembles to experimentally known structures, we performed a principal component analysis (PCA) on a set of all tubulin structures currently deposited in the PDB (*Figure 2*, *Figure 2—video 1*; see also Materials and methods). The PCA revealed the major conformational motion in this set and served as a measure of how intrinsic bending of the tubulin dimer in our simulations compared with that of experimental structures. *Figure 2A* shows all considered experimental structures on a two-dimensional (2D) plane spanned by the first two PCA conformational modes which describe the largest conformational variations among these structures (see also *Figure 2—figure supplement 1* for a simplified 2D sketch demonstrating the basic idea of this analysis). Each data point in this 2D plot represents one experimental structure, meaning that similarly kinked dimers are co-located along the x-axis. The PCA and visual inspection of all investigated structures indicate that these clearly fall into two subpopulations: straight tubulin or $TUB_S$ and kinked tubulin or $TUB_K$ (*Figure 2B*, green and pink points, respectively). Interestingly, no intermediate structures were observed in the known experimental data, suggesting that the population of such intermediates is probably very low and may not be resolved by cryo-EM and X-ray crystallography.

The first conformational mode contributes $\sim 90\%$ to the total structural variation in the set and describes a dimer bending motion (*Figure 2C*, *Figure 2—video 2*). Bending occurs around an 'anchor point' located at the intradimer interface that does not move during the transition. This anchor point involves hydrophobic interactions between the H8 helix of $\beta$-tubulin and the surface of $\alpha$-tubulin. Interestingly, a similar unaffected contact, but between adjacent dimers in MT protofilaments, has been previously shown to be involved in MT lattice compaction upon GTP hydrolysis (*Alushin et al., 2014*; *Zhang et al., 2015*). The second conformational mode contributes $\sim 5\%$ to the total structural variation and describes a 'breathing' motion of the dimer (*Figure 2—video 3*), which might be indicative of scaling differences in the experimental structures and/or different protein environment, particularly in those derived from cryo-EM reconstructions of MTs and 2D sheets of crystallized tubulin (*Figure 2A*, 3JAT/3JAS *vs.* 1TUB/1JFF). Notably, this difference is not present in the $TUB_K$ subpopulation which is exclusively constituted by X-ray structures co-crystallized with small ligands, stathmin-like factors or darpin.

*Figure 3* compares the four simulated tubulin ensembles to the experimental structures in terms of the above PCA results (green and pink trajectories *vs.* black dots). Most importantly, the nucleotide state seemed to strongly affect tubulin bending flexibility. While the GDP-tubulin ensembles were on average more 'confined' during the simulation time, GTP-tubulin was seen to explore a broader range of curvatures (*Figure 3*, motion along conformational mode 1), with many conformations being more kinked than is evident from the experimental structures (*Figure 3*, left panel, pink ensemble). Interestingly, all simulated structures started from straight $TUB_S$ conformations (*i.e.* instantaneously removed from the MT lattice where they are stabilized by neighboring dimers) quickly adopted similarly kinked conformations within several nanoseconds (*Figure 3—figure supplement 1*, top inserts), which suggests that these straight conformations may be highly unrelaxed in solution. Moreover, none of the simulations started from $TUB_S$ fully converged onto the curved conformation of unpolymerized tubulin; and conversely, none of the simulations started from $TUB_K$

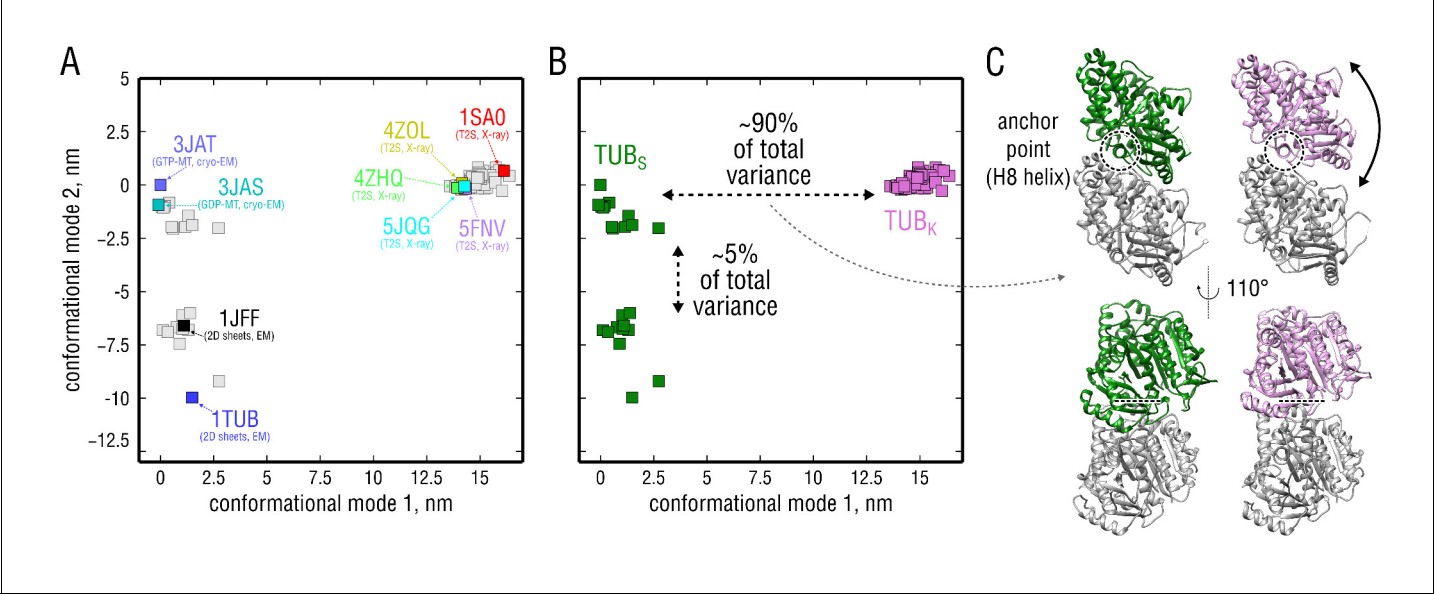

**Figure 2.** PCA on the set of experimental structures. (**A**) Projection of the set onto the plane constituted by the first and second largest-amplitude conformational modes. Each structure is represented by a point on this plane. Known experimental structures involved in this study (PDB ID: 3JAT, 3JAS, 5JQG, 5FNV, 4ZOL, 4ZHQ) as well as other structures widely referenced in literature (PDB ID: 1TUB, 1JFF, 1SA0) are highlighted. (**B**) Same data as in (**A**) but with straight (TUB$_S$) and kinked (TUB$_K$) subpopulations highlighted in green and pink, respectively. (**C**) Curvature differences between the subpopulations in (**B**) displayed in terms of structure. Shown are the extreme conformations along conformational mode 1 (x-axis). $\alpha$-tubulin is shown in gray and $\beta$-tubulin in green or pink, depending on the subpopulation. The anchor point that does not move during the straight-to-kinked transition is marked with a back dashed circle/line.

DOI: https://doi.org/10.7554/eLife.34353.003

The following video, source data, and figure supplement are available for figure 2:

**Source data 1.** Coordinates of the PDB structures projected onto conformational modes 1 and 2.
DOI: https://doi.org/10.7554/eLife.34353.005

**Figure supplement 1.** A simplified 2D example of the principal component analysis (PCA).
DOI: https://doi.org/10.7554/eLife.34353.004

**Figure 2—video 1.** Merged set of the analyzed PDB structures played in consecutive order.
DOI: https://doi.org/10.7554/eLife.34353.006

**Figure 2—video 2.** Structural motion along conformational mode 1.
DOI: https://doi.org/10.7554/eLife.34353.007

**Figure 2—video 3.** Structural motion along conformational mode 2.
DOI: https://doi.org/10.7554/eLife.34353.008

reached MT-like conformations within several microseconds. There are, we think, two possible explanations for the observed lack of connection between the TUB$_S$ and TUB$_K$ simulations: (**a**) insufficient sampling of one global state corresponding to the kinked dimer (**Figure 3**, hypothetically located at ~15 nm (GTP) and ~10 nm (GDP) along the x-axis), or (**b**) the presence of two global states of tubulin dynamics separated by a high free energy barrier (**Figure 3**, minima hypothetically placed at ~10 nm and ~17 nm (GTP) and ~7 nm and ~15 nm (GDP) along the x-axis). As shown in **Figure 3—figure supplement 1** (main four panels), independent equivalent simulations started from different experimental structures are locally converged, which motivated us to have a closer look at whether there was a hidden energy barrier separating the two sets of simulations in **Figure 3**.

## Trajectory analysis reveals two states with different bending modes

To test the possibility of two global states and an associated free energy barrier, we systematically derived a reaction coordinate (RC) that simultaneously takes into account both the experimental structures (**Figure 2A**) and our simulated ensembles (**Figure 3**), and reveals the highest barrier the dimer has to cross in order to transit from one state to the other (Materials and methods; see also **Voß, 2014**). This coordinate is termed the *ensemble separation* RC, and the main idea behind its

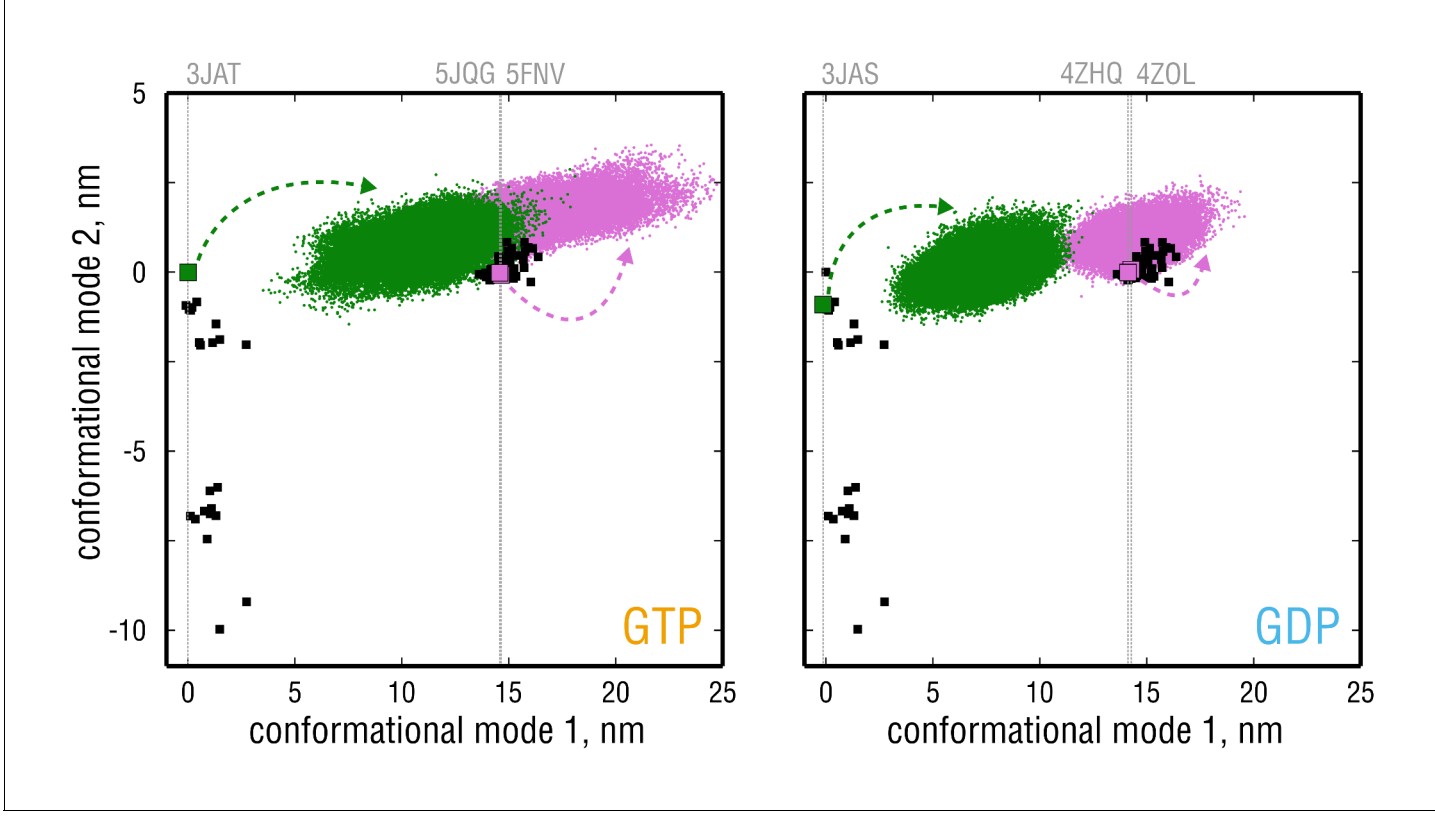

**Figure 3.** Simulated tubulin ensembles in two nucleotide states (left vs.right panel) plotted as in *Figure 2A,B*. Experimental structures are shown as black squares, whereas the simulated ensembles are shown as green (started from straight conformation) or pink (started from kinked conformation) point clouds. Experimental structures that were used for simulations are indicated with larger colored squares and vertical gray lines.
DOI: https://doi.org/10.7554/eLife.34353.009

The following source data and figure supplement are available for figure 3:

**Source data 1.** Coordinates of the merged GTP-tubulin ensemble started from a straight structure (PDB ID: 3JAT, two independent simulations) projected onto conformational modes 1 and 2 (left panel, green).
DOI: https://doi.org/10.7554/eLife.34353.011

**Source data 2.** Coordinates of the merged GTP-tubulin ensemble started from kinked structures (PDB IDs: 5JQG, 5FNV) projected onto conformational modes 1 and 2 (left panel, pink).
DOI: https://doi.org/10.7554/eLife.34353.012

**Source data 3.** Coordinates of the merged GDP-tubulin ensemble started from a straight structure (PDB ID: 3JAS, two independent simulations) projected onto conformational modes 1 and 2 (right panel, green).
DOI: https://doi.org/10.7554/eLife.34353.013

**Source data 4.** Coordinates of the merged GDP-tubulin ensemble started from kinked structures (PDB IDs: 4ZOL, 4ZHQ) projected onto conformational modes 1 and 2 (right panel, pink).
DOI: https://doi.org/10.7554/eLife.34353.014

**Figure supplement 1.** Convergence of the simulated ensembles assessed by overlaying point clouds originating from different independent simulations (two independent simulations per bound nucleotide per curvature state).
DOI: https://doi.org/10.7554/eLife.34353.010

derivation is sketched in *Figure 4—figure supplement 1* for a simple 2D case. We note that, strictly speaking, the nature of the hypothesized barrier in the simulated ensembles may not necessarily be related to tubulin bending because conformational modes 1 and 2 in *Figure 3* were derived using only a limited set of experimentally known conformations.

A surprising result of this analysis is that the conformational change associated with the identified ensemble separation RC, indeed, was unrelated to tubulin bending or nucleotide state but rather involved rearrangements in the individual monomers and at the intradimer interface (*Figure 4— video 1*; discussed in detail in the last Results section). This fact is shown in *Figure 4* as 2D free

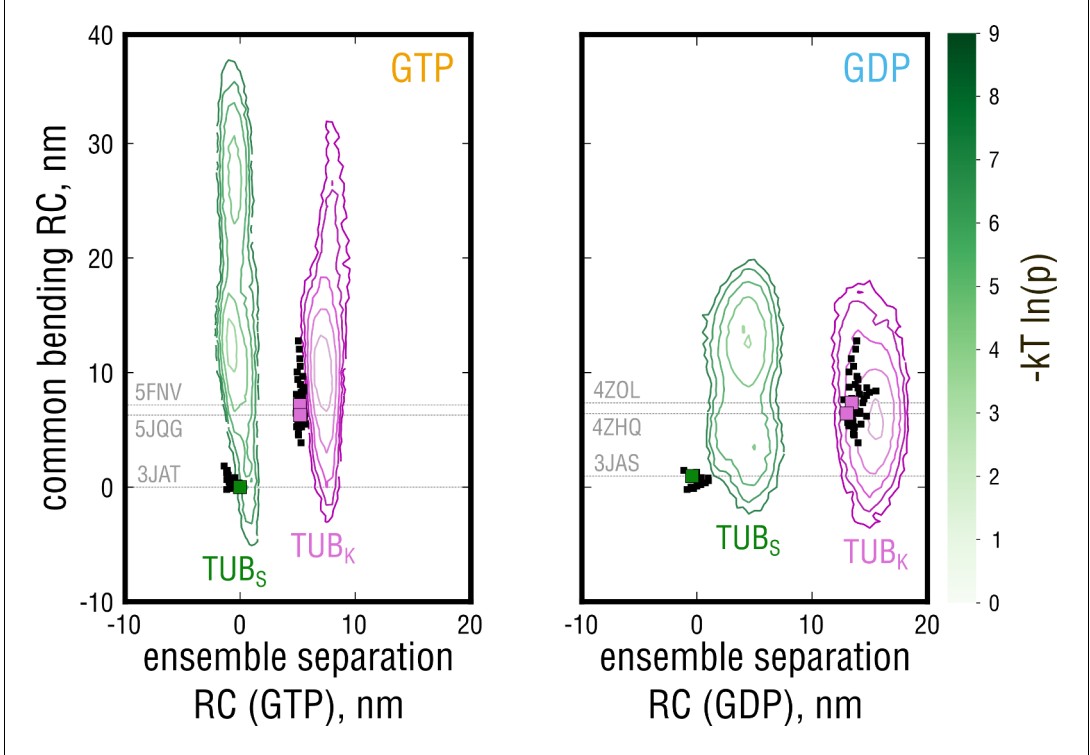

**Figure 4.** Free energy profiles derived from the projections of the simulated ensembles onto a plane, where the x-axis denotes the position along the ensemble separation RC and the y-axis denotes an orthogonal coordinate describing the largest-amplitude bending motion in the combined $TUB_S$ + $TUB_K$ ensemble. Experimental structures used for the free simulations are highlighted with gray dashed lines and green (straight structures) and pink squares (kinked structures).

DOI: https://doi.org/10.7554/eLife.34353.015

The following video, source data, and figure supplements are available for figure 4:

**Source data 1.** Coordinates of the PDB structures projected onto the ensemble separation and common bending coordinates (GTP).

DOI: https://doi.org/10.7554/eLife.34353.018

**Source data 2.** Coordinates of the merged GTP-tubulin ensemble started from a straight structure (PDB ID: 3JAT, two independent simulations) projected onto the ensemble separation and common bending coordinates (GTP).

DOI: https://doi.org/10.7554/eLife.34353.019

**Source data 3.** Coordinates of the merged GTP-tubulin ensemble started from kinked structures (PDB ID: 5JQG, 5FNV) projected onto the ensemble separation and common bending coordinates (GTP).

DOI: https://doi.org/10.7554/eLife.34353.020

**Source data 4.** Coordinates of the PDB structures projected onto the ensemble separation and common bending coordinates (GDP).

DOI: https://doi.org/10.7554/eLife.34353.021

**Source data 5.** Coordinates of the merged GDP-tubulin ensemble started from a straight structure (PDB ID: 3JAS, two independent simulations) projected onto the ensemble separation and common bending coordinates (GDP).

DOI: https://doi.org/10.7554/eLife.34353.022

**Source data 6.** Coordinates of the merged GDP-tubulin ensemble started from kinked structures (PDB ID: 4ZOL, 4ZHQ) projected onto the ensemble separation and common bending coordinates (GDP).

DOI: https://doi.org/10.7554/eLife.34353.023

**Figure supplement 1.** A 2D example of the ensemble separation search.

DOI: https://doi.org/10.7554/eLife.34353.016

**Figure supplement 2.** Dot products of the three-dimensional ensemble separation vector with vectors of higher dimensions for the tubulin ensembles in two different nucleotide states.

DOI: https://doi.org/10.7554/eLife.34353.017

**Figure 4—video 1.** Structural motion along the ensemble separation RC.

DOI: https://doi.org/10.7554/eLife.34353.024

energy profiles along the ensemble separation RC and a common bending RC. This bending coordinate was defined by performing a PCA on the sum of both simulated ensembles as was done for the experimental structures in *Figure 2*. First, *Figure 4* clearly shows no overlap between the $TUB_S$ and $TUB_K$ simulated ensembles along the ensemble separation RC, confirming the lack of overlap in *Figure 3*. Second, the motion along the ensemble separation RC is uncoupled from the bending motion both for GTP- and GDP-tubulin, confirming that the identified free energy barrier is unrelated to both tubulin curvature and nucleotide state. Finally, it can be inferred from these 2D profiles that, in our simulations, tubulin easily probes a wide range of curvatures (motion along the y-axis), independently of the starting conformation and with GDP-tubulin sampling a much more restricted range of curvatures (*Figure 4*, left *vs.* right panel).

We assume the reason for the poor ensemble overlap in *Figure 3* is likely the exclusive use of experimental structures to deduce the bending transition (*Figure 2*), which is less informative and in part masks the presence of the high free energy barrier identified by our ensemble separation search. Hence, the combination of experimental data and simulation yields a more complete view on the bending transition and reveals an additional conformational change in unassembled tubulin (*Figure 4—video 1*) – to our knowledge so far unknown – that is observed both in experimental and simulated structures.

## Tubulin gains flexibility upon GTP binding independently of the bending mode

We next focused on a quantitative characterization of the nucleotide-dependent bending motions (*Figure 4*, vertical axis). The fact that we observed two global free energy minima for tubulin in both nucleotide states suggests that the bending dynamics may be different in either of these. To test this possibility, we treated both minima separately. *Figure 5A* shows molecular representations of the largest-amplitude bending motions of GTP-tubulin in both free energy minima (see also *Figure 5—video 1* and Materials and methods for details on the derivation of these motions). Although projections of these motions onto the corresponding 2D free energy profile look very similar (*Figure 5B*, green and pink lines), both considerably differ in the way the tubulin dimer bends. While the motion derived from the $TUB_K^{GTP}$ ensemble (pink mode) mainly describes bending 'orthogonal' to the MT wall, resembling protofilament splaying at the MT end, the motion derived from the $TUB_S^{GTP}$ ensemble (green mode) includes a 'tangential' component, *i.e.* twisting of the $\beta$-subunit relative to the $\alpha$-subunit around the imaginary MT axis. To distinguish between these two different motions later on, we refer to them as the *splay-bend* (SB) and *twist-bend* (TB) mode, respectively.

Having identified the SB and TB modes of tubulin bending enabled us to directly test the lattice and allosteric models by calculating free energy profiles along these modes (*Figure 5B*) and to assess which of the two scenarios is more feasible from the energetic point of view. According to the lattice model (*Figure 1*, left branch), one would expect the profiles for GTP- and GDP-tubulin to be very similar, with kinked conformations having lower free energy values in either case. In contrast, the allosteric model (*Figure 1*, right branch) predicts that straight conformations of GTP-tubulin would have lower free energy values than kinked conformations, with the opposite being true for GDP-tubulin. Both lattice and allosteric models are unsupported by our free energy calculations, which suggests a different mechanism of allosteric control through GTP binding by tubulin.

Comparison of the free energy profiles for GTP- and GDP-tubulin revealed that GTP-tubulin favors a much broader range of intrinsic curvatures, all of which are almost isoenergetic (*Figure 6* and *Figure 6—figure supplement 1*; free energy differences of $1 - 2\,kT$). This observation is valid both for TB and SB motion. Contrary to what is postulated by the allosteric model, GTP does not shift the bending preference of tubulin toward low-curvature states according to our calculations. Rather, the range of curvatures energetically accessible to GDP-tubulin becomes strongly restrained (steeper free energy curves in *Figure 6*), while intermediate dimer kinking is preserved in either nucleotide state (approximate coincidence of the free energy minima in *Figure 6*). This suggests that tubulin might gain considerable flexibility through binding GTP without changing its average, intermediately kinked conformation.

We also estimated the free energy of kinking ($\Delta G_{kink}$) stored in GTP- and GDP-tubulin that are held in the straight, MT-like conformation. For this aim, the free energy values for the GTP- and GDP-tubulin structures derived from cryo-EM reconstructions of MTs and used in our free

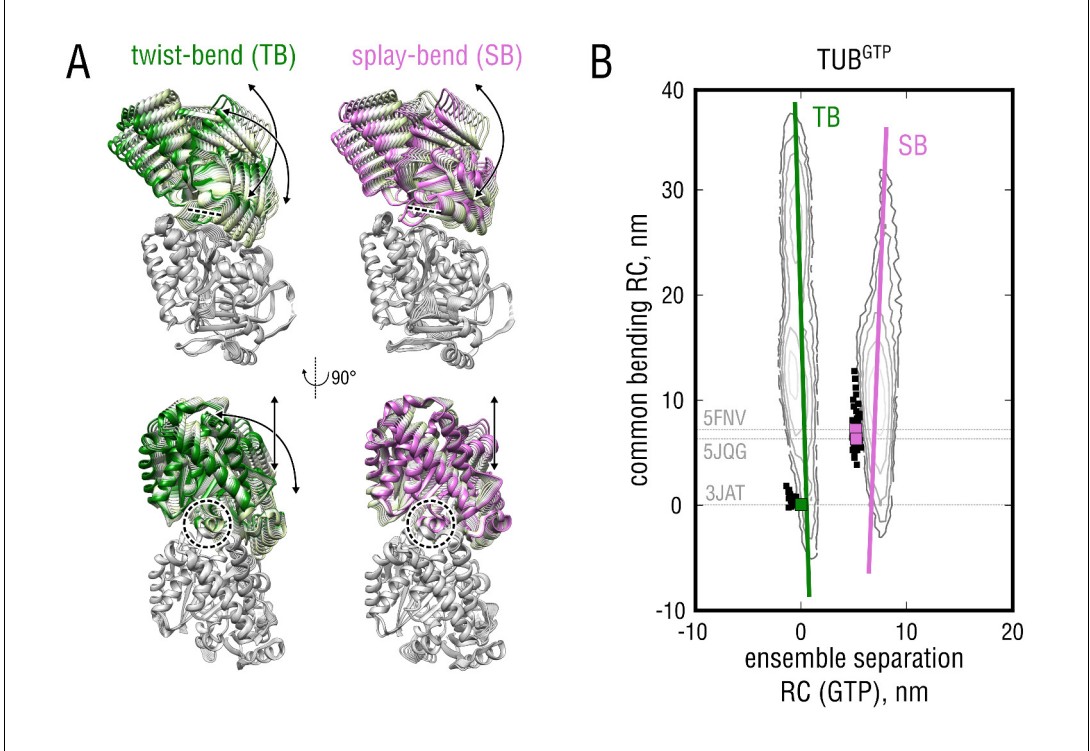

**Figure 5.** Nucleotide-dependent motions of tubulin bending. (A) Molecular representation of the twist-bend and splay-bend tubulin bending motions derived from the $TUB_S^{GTP}$ and $TUB_K^{GTP}$ simulated ensembles. The anchor point around the H8 helix is indicated with dashed lines/circles. See also *Figure 5—video 1* for a side-by-side comparison. (B) Same data as in *Figure 4* (left) with the derived bending motions shown as solid lines overlayed onto the free energy distributions. Experimental structures used for the free simulations are highlighted with gray dashed lines and green (straight structures) and pink squares (kinked structures).

DOI: https://doi.org/10.7554/eLife.34353.025

The following video and figure supplement are available for figure 5:

**Figure supplement 1.** Functional mode analysis of the $TUB_S^{GTP}$ and $TUB_K^{GTP}$ ensembles.
DOI: https://doi.org/10.7554/eLife.34353.026

**Figure 5—video 1.** Side-by-side comparison of the twist-bend (green) and splay-bend (pink) conformational motions.
DOI: https://doi.org/10.7554/eLife.34353.027

---

simulations (*Zhang et al., 2015*) were calculated (*Figure 6*, left panel, black and gray squares). Our estimations yield $\Delta G_{kink}^{GTP} = 2.0 \pm 0.4\,kT$ and $\Delta G_{kink}^{GDP} = 6.6 \pm 0.8\,kT$, suggesting that it costs less free energy for GTP-tubulin to adopt the straight MT conformation. Importantly, the calculated $\Delta G_{kink}^{GDP}$ is in agreement with the most recent estimate of curvature strain in straight GDP-protofilaments of $\sim 5\,kT$ per dimer obtained by laser tweezer measurements (*Driver et al., 2017*). We also emphasize that these energy costs do not include energetic contributions from neighboring dimers in the MT lattice.

## Tubulin has different preference for each of the bending modes

The free energy barrier identified by the ensemble separation search does not depend on tubulin curvature or nucleotide state (*Figure 4*). It separates the free energy basins with different bending dynamics in either of them (*Figure 5*; *Figure 6*). It is hence likely that this barrier, in addition to the nucleotide, controls tubulin's activity by making one or the other mode more favorable without interfering with the dimer curvature. We therefore analyzed the conformational change associated with the barrier and calculated the energetics of this transition.

Visual inspection of the respective collective motion (*Figure 7A* and *Figure 4—video 1*) revealed several large-amplitude internal rearrangements in the $\alpha$- and $\beta$-tubulin monomers that might contribute to the observed barrier. Those include an upward-downward movement of the central H7

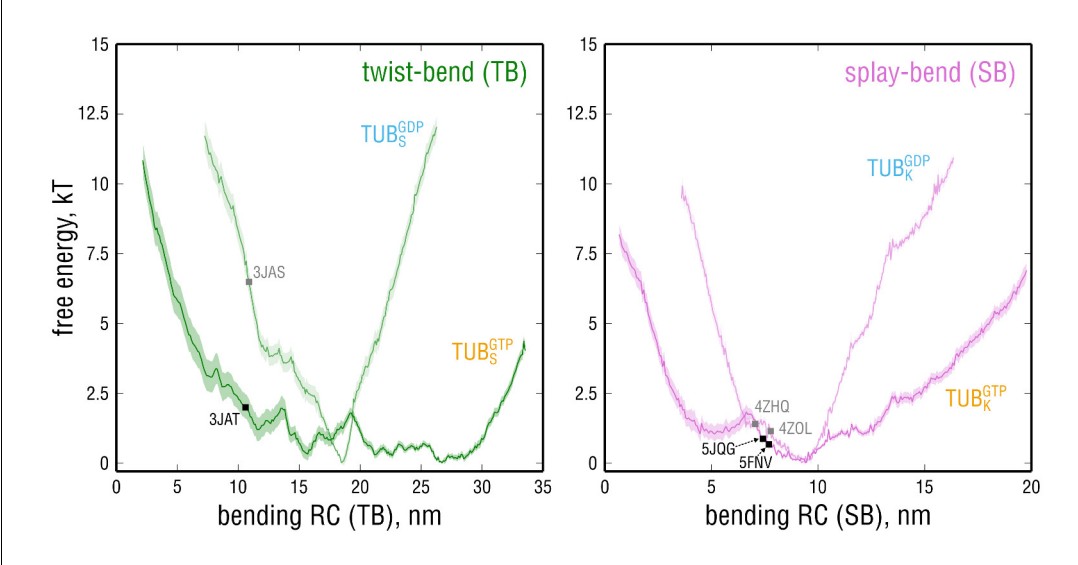

**Figure 6.** Free energy profiles along the bending RCs defined for the TB and SB bending motions calculated for GTP- and GDP-tubulin. Smaller (larger) values on the x-axis correspond to straighter (more kinked) dimer conformations. Squares denote the positions of the starting PDB structures in the GTP- (black) and GDP-state (gray) used for the free MD simulations. Note that the scaling for both RCs does not necessarily coincide with that of the common bending RC in *Figure 4*, as the vectors defining the TB and SB motions and the common bending motion were derived differently (see Materials and methods).

DOI: https://doi.org/10.7554/eLife.34353.028

The following source data and figure supplement are available for figure 6:

**Source data 1.** Free energy values (second and third columns, mean +/- SD) along the TB bending coordinate (first column) for GTP-tubulin.
DOI: https://doi.org/10.7554/eLife.34353.030
**Source data 2.** Free energy values (second and third columns, mean +/- SD) along the TB bending coordinate (first column) for GDP-tubulin.
DOI: https://doi.org/10.7554/eLife.34353.031
**Source data 3.** Free energy values (second and third columns, mean +/- SD) along the SB bending coordinate (first column) for GTP-tubulin.
DOI: https://doi.org/10.7554/eLife.34353.032
**Source data 4.** Free energy values (second and third columns, mean +/- SD) along the SB bending coordinate (first column) for GDP-tubulin.
DOI: https://doi.org/10.7554/eLife.34353.033
**Figure supplement 1.** Convergence properties and robustness of the error evaluation with respect to the method used.
DOI: https://doi.org/10.7554/eLife.34353.029

helix in the intermediate domain (residues 206–371) accompanied by a translocation of the H7-H8 loop, which leads to a shift in the position of the anchor point (motions of non-interacting flexible loops on the outer surface were not considered). Analysis of the same transitions in GDP-tubulin yielded a qualitatively similar picture of the intradimer rearrangement (not shown). Hence, crossing the barrier along the ensemble separation RC does not require the dimer to bend but rather induces changes in the monomer structure and at the intradimer interface.

It was not possible to estimate individual basin depths ($\Delta G_{TB}$ and $\Delta G_{SB}$) as well as the free energy difference between the two states ($\Delta\Delta G_{TB-SB} = \Delta G_{TB} - \Delta G_{SB}$) based on the free simulations (*Figure 4*), because the transition state was only sparsely populated. We therefore performed free energy calculations to overcome sampling issues in the transition region and to reconstruct the full free energy landscape along the ensemble separation RC (*Figure 7B* and *Figure 7—figure supplement 1*). The individual basin depths (relative to the transition state at $\sim 2.5$ nm) were found to be $\Delta G_{TB} = -11.5 \pm 0.5\,kT$ and $\Delta G_{SB} = -20.1 \pm 2.5\,kT$, thus confirming the absence of spontaneous transitions in the $\mu s$-long free simulations. Consequently, the free energy difference between the states was estimated to be $\Delta\Delta G_{TB-SB} = 8.6 \pm 2.4\,kT$, suggesting that the basin corresponding to the SB motion is energetically much more favorable ($\sim 99.98\%$ of the free GTP-tubulin population will exhibit the SB bending motion).

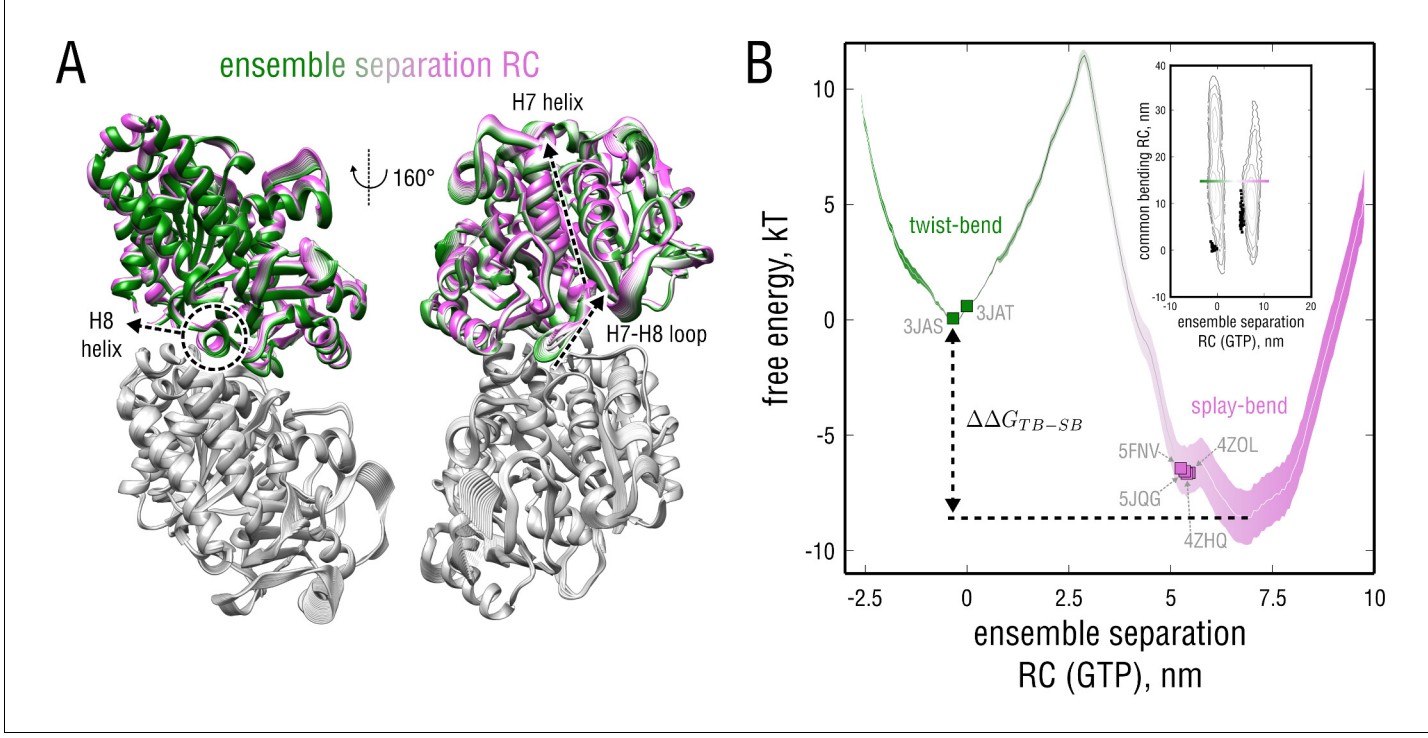

**Figure 7.** Dynamics and energetics of GTP-tubulin along the ensemble separation RC. (**A**) Collective mode of motion along the ensemble separation RC represented as a linear interpolation between the $TUB_S$ and $TUB_K$ simulated ensembles. The largest-amplitude intrinsic rearrangements in $\beta$-tubulin are indicated with dashed lines and involve the H7 and H8 helices as well as a loop connecting them. (**B**) Free energy landscape as a function of the ensemble separation RC. Orientation of this coordinate with respect to the TB and SB free energy basins is schematically shown as an insert. Experimental structures used for the free simulations are highlighted with green (straight structures) and pink squares (kinked structures).
DOI: https://doi.org/10.7554/eLife.34353.034

The following source data and figure supplement are available for figure 7:

**Source data 1.** Free energy values (second and third columns, mean +/- SD) along the ensemble separation coordinate (first column).
DOI: https://doi.org/10.7554/eLife.34353.036
**Figure supplement 1.** Convergence properties and robustness of the error evaluation with respect to the method used.
DOI: https://doi.org/10.7554/eLife.34353.035

## Discussion

Our results suggest a new mechanism by which GTP binding by unassembled tubulin is linked to its conformational dynamics. This mechanism reconciles the previously proposed lattice and allosteric models (*Figure 1*) into a new unified model, hence resolving the discrepancies between previous experimental observations. We propose that tubulin enters the MT lattice via a combination of both mechanisms (*Figure 8*). In solution, tubulin can exist in two anchor point states, each capable of nucleotide-dependent bending but differing in the way the dimer bends, with the SB bending mode being strongly energetically favored over the TB mode (*Figure 4*; *Figure 7B*; *Figure 8*, horizontal transition in the bottom cycle). The average dimer conformation in either state is intermediately kinked, irrespective of the bound nucleotide (*Figure 6*), which might explain experimental observations that colchicine binding and SAXS profiles of soluble GTP- and GDP-tubulin are almost identical (*Manuel Andreu et al., 1989*; *Rice et al., 2008*). In this respect, our findings are also consistent with the early dimer (*Gebremichael et al., 2008*) and follow-up protofilament simulations (*Grafmüller and Voth, 2011*; *Grafmüller et al., 2013*) where both dimers and protofilaments (GTP and GDP) attained largely kinked conformations on the several tens of nanoseconds timescale. We have now shown that bending flexibility at the intradimer interface is controlled by the nucleotide state and gets strongly enhanced upon GTP binding (*Figure 6*; *Figure 8*, upward transition in the bottom cycle). Although allosteric control through GTP binding is still present in our model, it differs

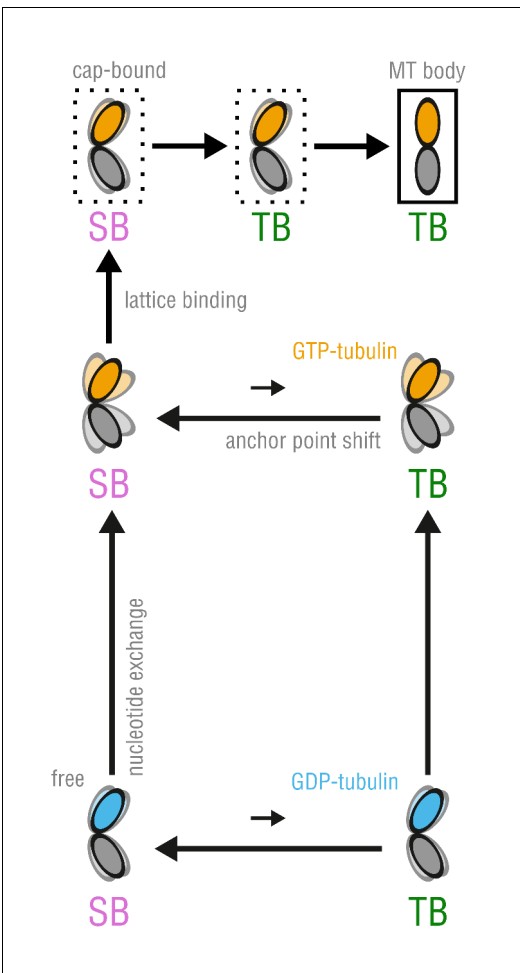

**Figure 8.** The proposed, combined model of MT assembly represented by a thermodynamic cycle. Here, TB and SB denote not only the different modes of tubulin bending but also the belonging to the respective free energy basin, *i.e.* anchor point state.

DOI: https://doi.org/10.7554/eLife.34353.037

conceptually from that proposed by the canonical allosteric model (**Figure 1**, right branch), wherein GTP binding forces tubulin to undergo a kinked-to-straight conformational change.

In our model, the GTP-induced increase in flexibility does not force tubulin to adopt the straight MT-like conformation, which is in part compatible with the lattice model (**Figure 1**, left branch). Straightening of kinked tubulin at the MT tip and within the lattice is accomplished through induced fit, *i.e.* the energy of kinking stress (**Figure 6**, left) is compensated by lateral and longitudinal bond energies. This compensation is merely a lattice effect and does not involve allosteric regulation via GTP. Notably, this scenario does not contradict the experimental data where MTs polymerized with a non-hydrolyzable GTP-analog, GMPCPP, were seen to depolymerize into significantly less curved protofilaments than normal MTs (**Müller-Reichert et al., 1998**) and intermediate-curvature states of polymers made of GMPCPP-tubulin were observed by cryo-EM (**Wang and Nogales, 2005**; **Nogales and Wang, 2006**). Formation of straight or less kinked GTP-tubulin conformations at MT tips, we think, might not strictly require that those conformations be energetically favored in solution. Holding GTP-tubulin in the straight lattice conformation is less challenging for the MT lattice, as evidenced by $\Delta G_{kink}$ calculated for GTP-tubulin ($\sim 2.0\,kT$) and GDP-tubulin ($\sim 6.6\,kT$) (**Figure 6**, left). Less kinked, cap-bound GTP-dimers would be more likely to be observed, given at least one lateral neighbor. Finally, the loaded-spring-like stress caused by $\Delta G_{kink}$ being non-zero both for GTP- and GDP-tubulin, together with the axial compaction in the lattice induced by GTP hydrolysis

(*Alushin et al., 2014*; *Zhang et al., 2015*), could generate destabilizing strain that would be released during MT depolymerization.

The coexistence of the two curvature- and nucleotide-independent anchor point states separated by a large free energy barrier is striking (*Figure 4*; *Figure 7*). It also raises the question how such a high barrier of $\sim 20\,kT$ in unassembled tubulin is compatible with its polymerization dynamics. We think there are two possible scenarios. The first possibility is sketched in *Figure 8* (top cycle) and assumes that the activation energy to cross the SB-TB barrier for a newly arrived dimer at the MT tip comes from favorable lateral and longitudinal bond formation. The second possibility is that SB and TB dimers are equally well incorporated into the MT lattice, and that the transition may be accomplished much later in the MT body. While we cannot discriminate between the two possibilities at present, we favor the first because: (a) no tubulin structure derived from the MT body has (yet) been deposited showing a dimer in the SB state; and (b) the values for longitudinal and lateral bond energies found in literature seem to be sufficient to antagonize the high SB-TB barrier (*VanBuren et al., 2002*; *VanBuren et al., 2005*; *Gardner et al., 2011*; *Castle and Odde, 2013*; *Kononova et al., 2014*). More specifically, the longitudinal bond energy ranges between $-16.8\,kT$ and $-25.2\,kT$ (not including the entropic cost of binding from the bulk to the MT tip which is $\sim 10\,kT$ ([*Castle and Odde, 2013*]), while the lateral bond energy is estimated to be between $-3.6\,kT$ and $-11.7\,kT$. Taking the lower bounds of these estimates, one arrives at a combined (one lateral and one longitudinal) energy of $-20.4$ kT, which is already enough to counterbalance the high SB-TB activation energy.

Although we cannot provide strong evidence for the functional role of the barrier separating the two anchor point states (*Figure 7*), it is tempting to speculate that this barrier, in addition to the nucleotide state, may control the kinetics of MT assembly and disassembly, *i.e.* MT catastrophe rates. In fact, this barrier might resolve the so far unexplained inability of modern mechanochemical computational models of MT assembly (*VanBuren et al., 2002*; *VanBuren et al., 2005*; *Margolin et al., 2011*; *Margolin et al., 2012*; *Castle and Odde, 2013*; *Zakharov et al., 2015*) to reproduce typical MT lifetimes (several minutes), lengths (several microns), as well as their moderate dependence on free tubulin concentration observed in vitro (*Walker et al., 1988*; *Walker et al., 1991*; *Gardner et al., 2011*), which are the key features of MT dynamic instability. This dependence of MT lifetimes and lengths on free tubulin concentration is predicted to be too steep (*VanBuren et al., 2002*), implying a characteristic time for a catastrophe event under physiologically relevant concentrations being on the order of years (*Zakharov et al., 2015*). There must be therefore a lack of important mechanical features which causes a 'hyperstabilization' of the MT in these models. Our results obtained with atomistic MD simulations may now provide some of these missing features.

In conclusion, our results, combined with previous structural knowledge, extend our understanding of MT dynamic instability. In particular, the new combined model of MT assembly cross-bridges previous contradictory experimental observations and may address one of the longstanding questions in the MT field, namely: why does GTP-tubulin polymerize and GDP-tubulin does not? As shown by our work, assuming that GTP-induced tubulin flexibility, and not the dimer conformation as such, is the driving force for MT assembly resolves this discrepancy and reconciles contradictory experimental data reported previously. Thus, we believe that this new mechanism is an important step toward revisiting our understanding of the MT life cycle and accounts for unexplained complexities of MT growth and catastrophe.

## Materials and methods

### Ensemble of experimental structures

The set of tubulin structures was extracted from the PDB using the BLAST method (*Altschul et al., 1990*). The template sequences of $\alpha$- and $\beta$-tubulin were extracted from the structure of a straight GMPCPP-tubulin dimer (PDB ID: 3JAT [*Zhang et al., 2015*]). At the time of performing the search (April, 2017), the PDB contained $\sim 100$ tubulin structures sharing at least 80% sequence similarity with the template sequences (*sus scrofa*). This preliminary set was then aligned by filtering out structures whose chains shared less than 96% sequence identity and whose aligned parts of the sequences covered less than 85% of the template sequence using ProDy (*Bakan et al., 2014*), *i.e.* structures

that deviated strongly from the reference (3JAT) in terms of sequence or that had too many missing residues were excluded. This additional filtering yielded a set with a total of 91 structures. This final set covered a broad range of tubulin structures: 62 structure in the kinked conformation, 29 structures in the straight conformation as well as various nucleotide states (GTP, GDP, GTP-analogs like GMPCPP etc.). Most of the kinked structures were in the so-called T2S complex with the MT depolymerizing factor Op18/stathmin (*Belmont and Mitchison, 1996*), whereas all straight structures originated from cryo-EM reconstructions of MTs or 2D sheets of crystallized tubulin. Residue numbering mismatches in the β-subunits were fixed. All residues of β-tubulin are referred to in the main text according to the corrected numbering.

## Principal component analysis

Principal component analysis (PCA) (*Amadei et al., 1993*; *de Groot et al., 1996*) was performed on structural sets/ensembles using only the backbone atoms and excluding the flexible C-termini. To clarify the nomenclature, we briefly describe the essence of the PCA below. Given a set of $n$ atomic configurations (e.g., a set of PDB structures or a MD trajectory), each consisting of $N$ atoms $\{\mathbf{x}_k \in R^{3N}\}$, the main goal of a PCA is to reduce the dimensionality of the conformational space by finding $d$ unit vectors in $R^{3N}$ ($d \ll 3N$) which describe most of the ensemble variance. For this aim, after removing translational and rotational motions by alignment of the ensemble with a reference structure $\mathbf{x}_{ref}$ (here, PDB ID: 3JAT), the covariance matrix of atomic positions $\mathbf{C}$ is constructed,

$$C_{ij} = \langle (x_i - \langle x_i \rangle)(x_j - \langle x_i \rangle) \rangle, \tag{1}$$

where $\mathbf{x}$ represents the average configuration. Diagonalization of $\mathbf{C}$ yields a set of $3N$ orthogonal unit eigenvectors $\mathbf{v}_k$ and eigenvalues $\lambda_k$ (in descending order),

$$\mathbf{C} = \mathbf{Q}\mathbf{\Lambda}\mathbf{Q}^T = (\mathbf{v}_1, \dots, \mathbf{v}_{3N}) \begin{pmatrix} \lambda_1 & \cdots & 0 \\ \vdots & \ddots & \vdots \\ 0 & \cdots & \lambda_{3N} \end{pmatrix} (\mathbf{v}_1, \dots, \mathbf{v}_{3N})^T. \tag{2}$$

The projection of the ensemble onto the $k^{th}$ eigenvector, $q_k = (\mathbf{x} - \langle \mathbf{x} \rangle) \cdot \mathbf{v}_k$, is termed the $k^{th}$ principal component or conformational mode and denotes the collective motion along this eigenvector with the variance given by $\lambda_k$. For molecular simulations, the first 1–20 conformational modes usually account for 80–90% of the ensemble variance (*Amadei et al., 1993*).

## MD simulations

Six tubulin structures were selected for subsequent MD simulations: two straight dimers (PDB IDs: 3JAT (GMPCPP), 3JAS (GDP)) and four kinked dimers (PDB IDs: 5JQG (GTP), 5FNV (GTP), 4ZHQ (GDP), 4ZOL(GDP)). For convenience, we refer to the straight structures as $\text{TUB}_S^{GTP}$ and $\text{TUB}_S^{GDP}$, and to the kinked structures as $\text{TUB}_K^{GTP}$ and $\text{TUB}_K^{GDP}$. Missing atoms and residues as well as the C-termini ($\alpha$:437–451, $\beta$:426–445; presumably unstructured) were added using MODELLER version 9.17 (*Fiser et al., 2000*). Protonation states of the histidines were assigned using the GMCT package (*Ullmann and Ullmann, 2012*). The straight tubulin structures were extracted from $3 \times 2$ lattice patch models (3JAT, 3JAS) of which one (3JAT) contained the non-hydrolyzable GTP analog (GMPCPP) in the E-site of β-tubulin. GMPCPP was converted into GTP by replacing the carbon atom between $\alpha$- and $\beta$-phosphate with an oxygen atom, and the new bond lengths and angle relaxed to their equilibrium values during minimization. The kinked structures were extracted from T2S complexes coassembled with multiple small ligands (calcium and chloride ions, ethanesulfonic acid, glycerol, etc.). Stathmin as well as the small ligands were stripped out while preserving the GDP and $Mg^{2+}$-coordinated GTP molecules.

GROMACS version 4.6 (*Szilárd et al., 2015*) and CHARMM22* force field (*Piana et al., 2011*) were used for all simulations. The simulation setup was similar to that described in (*Rauscher et al., 2015*). Briefly, for every nucleotide/curvature state $\text{TUB}_X^Y$, the simulated system consisted of a tubulin dimer centered in a rhombic dodecahedral box filled with CHARMM-modified TIP3P water (*MacKerell et al., 1998*) and 0.15M KCl ($\sim 170,000$ atoms in total). Center-of-mass subtraction was

applied to the solute's atoms (including GTP, GDP, and $Mg^{2+}$ in GTP) in order to prevent the dimer from drifting away from the periodic box, which permitted more convenient trajectory analysis.

Prior to the production runs, each system was subject to an initial equilibration phase involving steepest-descent energy minimization followed by 1 ns of MD simulation in the NVT ensemble at 100K with position restraints applied to the solute's non-hydrogen atoms (only protein) and 6 ns of simulation in the NPT ensemble at 1 atm during which the temperature was gradually increased from 100K to 300K. The LINCS algorithm (*Hess et al., 2008*) was used to constrain the lengths of bonds with hydrogen atoms, allowing a 2 fs integration time step. The cutoff radius for the Lennard-Jones and short-range electrostatic interactions was set to 0.95 nm, and long-range electrostatics were calculated using the particle-mesh Ewald (PME) method (*Essmann et al., 1995*) with a 0.12 nm grid spacing. The velocity rescaling thermostat (*Bussi et al., 2007*) was used for all simulations. Berendsen pressure coupling (*Berendsen et al., 1984*) was used to maintain the atmospheric pressure during equilibration.

For each $TUB_X^Y$, the last frame of the equilibration phase was extracted, and virtual sites (*Hess et al., 2008*) for the solute were introduced to remove the fastest degrees of freedom. Together with constraining all bond lengths with the LINCS algorithm, this allowed a 4 fs integration time step. The simulation was then continued in the NPT ensemble at 300K using the Parrinello-Rahman algorithm (*Parrinello and Rahman, 1981*) for pressure control for a total duration of $3\,\mu s$. For $TUB_S^{GTP}$ and $TUB_S^{GDP}$, two independent production runs per system were conducted and combined. For $TUB_K^{GTP}$ and $TUB_K^{GDP}$, production runs started from two different crystal structures were combined. This yielded an accumulated trajectory of $6\,\mu s$ for each $TUB_X^Y$, covering a total of $24\,\mu s$ of free tubulin dynamics. Source files and a step-by-step guide necessary to reproduce the simulations as described above are freely available (https://github.com/moozzz/simulation-protocols/tree/master/free-tubulin-simulation [*Igaev, 2018a*]; copy archived at https://github.com/elifesciences-publications/simulation-protocols). Unless differently specified, VMD (*Humphrey et al., 1996*) and UCSF Chimera (*Pettersen et al., 2004*) were used for visualization and native GROMACS tools for structure analysis.

## Ensemble separation search

To study the energetics of and transitions between different conformational states, the ensemble separation search was employed (*Voß, 2014*), yielding a unit vector $\mathbf{n} \in R^{3N}$ that best separates two minima in the multidimensional free energy landscape. Essentially, given two simulated ensembles $\mathbf{x}_{1,2}(t)$ projected onto a PCA-based vector space, the method seeks the best linear reaction coordinate such that the projections of the probability densities of the two ensembles $\rho_{1,2}(\mathbf{x}) \in R^{3N}$ onto this vector have the smallest overlap. The optimization problem consists in finding the minimum of the following overlap integral:

$$O(\mathbf{n}) = \int \bar{\rho}_1(z, \mathbf{n})\bar{\rho}_2(z, \mathbf{n})dz, \qquad (3)$$

where $\bar{\rho}_{1,2}$ are projections of the multidimensional probability densities $\rho_{1,2}$ sampled by the two ensembles onto $\mathbf{n}$:

$$\bar{\rho}_{1,2}(z, \mathbf{n}) = \int \rho_{1,2}(\mathbf{x})\delta(\mathbf{n} \cdot \mathbf{x} - z)d^{3N}\mathbf{x}, \qquad (4)$$

and $z$ represents the reaction coordinate along the axis defined by $\mathbf{n}$.

In practice, a search in the $3N$-dimensional conformational space is unfeasible given a molecular system with thousands of atoms due to the curse of dimensionality. Using the basic property of the PCA, that is, dimensionality reduction, it is however possible to accelerate the search by approximating $\mathbf{n}$ with a linear combination of only the first $d \ll 3N$ PCA eigenvectors. Here, $d$ serves as a regularization parameter which is incremented until the solution to *Equation 4* becomes independent of $d$.

The ensemble separation RC search algorithm was implemented in a custom-made script (https://github.com/moozzz/orc_search_gauss [*Igaev, 2018b*]; copy archived at https://github.com/elifesciences-publications/orc_search_gauss). For a given $d$, the search consisted of two major steps. First, $10^6$ random unit vectors were generated (500 scans $\times 2000$ vectors) with the coordinates being

drawn from $d$ independent, one-dimensional normal distributions $N_i(\mu = 0, \sigma^2 = 1)$. In each scan, the vector that corresponded to the minimal overlap in *Equation 4* was selected, and a local, gradient-free minimization was carried out by the downhill simplex method (*Nelder and Mead, 1965*). The vectors resulting from the independent scans were then averaged to yield the final ensemble separation vector $\mathbf{n}$. Convergence of the search was assessed by monitoring the variances of individual coordinates of $\mathbf{n}$ as functions of the number of scans.

A new vector basis was first derived by a PCA on the combined $12\,\mu s$ $TUB_S$ and $TUB_K$ trajectory for each nucleotide state. The first $d$ PCA eigenvectors were then used as a basis set to find the ensemble separation vector $\mathbf{n}$ that minimized the overlap between the two ensembles, where $d$ was varied to account for all relevant motions in the ensembles. In both cases, less than 20 PCA eigenvectors were sufficient to separate the ensembles (*Figure 4—figure supplement 2*). Finally, the ensembles were projected onto a 2D plane constituted by the ensemble separation vector and an orthogonal vector with the largest variance in the combined ensemble (*i.e.* the first PCA eigenvector in the $(d - 1)$-dimensional subspace; *Figure 4*).

## Deriving bending modes with functional mode analysis

Functional mode analysis (FMA) (*Hub and de Groot, 2009*; *Krivobokova et al., 2012*) was applied to derive the collective motion of protein dynamics related to a particular functional quantity – here, tubulin bending. In its partial least-squares (PLS) form (*Krivobokova et al., 2012*), the FMA yields a vector basis with the lowest possible dimensionality $k \ll 3N$ that guarantees optimal correlation between the fluctuations in the functional quantity of interest and free protein dynamics. Briefly, in the PLS-based FMA, a regression problem of the form

$$\mathbf{f} = \mathbf{X}^T \mathbf{W}_k \beta_k + \epsilon \tag{5}$$

is solved, where $\mathbf{f}$ denotes the functional quantity of interest $f(t)$ represented as a vector, $\mathbf{X} = (\mathbf{x}_1, \mathbf{x}_2, \ldots, \mathbf{x}_n)$ is the $3N \times n$ matrix of atomic positions for an ensemble consisting of $n$ structures, $\mathbf{W}_k$ is a matrix composed of $k$ basis column vectors, and $\beta_k$ is a vector of coefficients to be optimized. The basis $\mathbf{W}_k$ is defined iteratively according to both variance in $\mathbf{X}$ (similarly to a PCA; see above) and linear correlation between $\mathbf{f}$ and $\mathbf{X}$. Like $d$ in the ensemble separation search, $k$ serves as a regularization parameter which is incremented until the maximal predictive power of the regression model in *Equation 5* is reached. This is achieved by simultaneous cross-validation against an independent set of data. Finally, an ensemble-weighted mode of motion is constructed that correlates best with $\mathbf{f}$ and has sufficient variance to contribute to $\mathbf{X}$. Details on the GROMACS implementation of the PLS-based FMA are given elsewhere (*Krivobokova et al., 2012*).

We chose the RMSD relative to the straight tubulin structure derived from the MT lattice (3JAT) as the functional quantity of interest (*Figure 7—figure supplement 1A*). Any deviation from the straight conformation increases the RMSD, irrespective of the bending direction. However, since the RMSD is a nonlinear function of the atomic coordinates, nonlinear correlation contributions cannot be assessed. Nevertheless, an acceptable quality model was obtained for this nonlinear case as shown in *Figure 7—figure supplement 1B,C*. Correlation coefficients between regression model and data are shown in *Figure 7—figure supplement 1B* for model building ($R_m$) and cross-validation ($R_v$) as a function of the FMA basis dimension for both $TUB_S^{GTP}$ and $TUB_K^{GTP}$ half-ensembles (from 0 to $3\,\mu s$). The cross-validation showed that the FMA converges at $\sim 8$ and $\sim 5$ components for the $TUB_S^{GTP}$ and $TUB_K^{GTP}$ ensembles, respectively. *Figure 7—figure supplement 1C* demonstrates the overlayed RMSD data half-set and the FMA fit shown for both model building and cross-validation parts, using the optimal number of components in each case. High correlation coefficients observed for model building and cross-validation in both cases ($\geq 0.95$) confirmed that the obtained FMA models are adequate and reflect the general features of RMSD fluctuations. The second $TUB_S^{GTP}$ and $TUB_K^{GTP}$ half-ensembles (from 3 to $6\,\mu s$) were used as additional and independent cross-validation sets, which yielded $R_v \approx 0.75$ and $R_v \approx 0.85$, respectively. Hence, the found FMA bending modes are a robust representation of the relation between the RMSD to the straight reference structure and the ensemble dynamics.

## Free energy calculations

To calculate the free energy profiles along the ensemble separation RC and the reaction coordinates defined by the FMA bending modes, umbrella sampling simulations were carried out (*Torrie and Valleau, 1977*; *Kästner, 2011*). In each case, starting configurations (seeds) were derived by projecting a free simulation trajectory onto a FMA bending mode or the ensemble separation RC and selecting structures placed along the bending coordinate with a step of $0.1 - 1$ nm. For those simulations, where the projected trajectory did not cover the desired range along the bending coordinate, additional seeds were generated from preceding umbrella windows using the structures closest to the desired value along the reaction coordinate. Harmonic potentials $V_i = \frac{1}{2}k(\mathbf{a} \cdot (\mathbf{x} - \mathbf{x}_i))^2$ with spring constants $k = 45 - 150\,kJ/mol$ were used to restrain the seed configurations, where $\mathbf{a}$ is the vector defining the reaction coordinate. For technical reasons, inverted flooding potentials were employed to approximate $V_i$ (*Lange et al., 2006*). Each of the restrained structures was simulated for 120–200 ns, and the first five ns in each simulation were discarded as equilibration. The free energy profile was reconstructed using the WHAM method (*Kumar et al., 1992*) implemented in the GROMACS g_wham tool, and its uncertainty was assessed by Bayesian bootstrapping of complete probability histograms (*Hub et al., 2010*).

As a further control, a conceptually unrelated method proposed by (*Zhu and Hummer, 2012*) was used to crosscheck the uncertainty of the free energy profiles. Unlike in Bayesian bootstrapping, where new sets of histograms are generated based on the sampling in each of the umbrella windows, this method relies on the statistical error of the mean force in every individual window. The mean force errors, in turn, are obtained from block averages, with the optimal block size calculated as $T_i/(2\tau_i + 1)$, where $T_i$ is the total duration of the trajectory and $\tau_i$ is the autocorrelation time of $\mathbf{a} \cdot \mathbf{x}(t)$ in window $i$.

## Acknowledgements

This work was supported by the Max Planck Society and the German Research Foundation (DFG) via the research grant No. IG 109/1–1. The authors gratefully acknowledge computer time provided by the North-German Supercomputing Alliance (HLRN). The authors also thank Gregory Bubnis, Nicholas Leioatts, Thomas Ullmann, and Andrea Vaiana for helpful discussions.

## Additional information

### Funding

| Funder | Grant reference number | Author |
|---|---|---|
| Max-Planck-Gesellschaft | Open-access funding | Maxim Igaev<br>Helmut Grubmüller |
| Deutsche Forschungsgemeinschaft | IG 109/1-1 | Maxim Igaev |

The funders had no role in study design, data collection and interpretation, or the decision to submit the work for publication.

### Author contributions

Maxim Igaev, Conceptualization, Data curation, Formal analysis, Funding acquisition, Investigation, Visualization, Writing—original draft, Project administration; Helmut Grubmüller, Conceptualization, Resources, Supervision, Funding acquisition, Writing—review and editing

### Author ORCIDs

Maxim Igaev [iD] http://orcid.org/0000-0001-8781-1604
Helmut Grubmüller [iD] https://orcid.org/0000-0002-3270-3144

### Decision letter and Author response

Decision letter https://doi.org/10.7554/eLife.34353.040

Author response https://doi.org/10.7554/eLife.34353.041

## Additional files

### Supplementary files
• Transparent reporting form
DOI: https://doi.org/10.7554/eLife.34353.038

### Data availability
A step-by-step guide for reproducing the simulations and all custom-made scripts used in this study have been uploaded to a publicly available GitHub repository (https://github.com/moozzz/simulation-protocols/tree/master/free-tubulin-simulation).

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
