## [Decision Letter]

Thank you for submitting your article "Microtubule assembly governed by tubulin allosteric gain in flexibility and lattice induced fit" for consideration by *eLife*. Your article has been reviewed by three peer reviewers, and the evaluation has been overseen by a Guest Reviewing Editor and Anna Akhmanova as the Senior Editor. The following individuals involved in review of your submission have agreed to reveal their identity: William O Hancock (Reviewer #2); Gregory A Voth (Reviewer #3).

The reviewers have discussed the reviews with one another and the Reviewing Editor has drafted this decision to help you prepare a revised submission.

Summary:

The manuscript from Igaev and Grubmuller uses molecular dynamics (MD) simulations to investigate whether/how the identity of the bound nucleotide (GTP vs GDP) affects the conformation of ab-tubulin. This is an interesting and challenging question that has not been definitively settled, and where new insights can impact the understanding of microtubule dynamics and regulation. The authors perform atomistic molecular dynamic (MD) simulations of GTP/GDP-tubulin dimers starting from different conformations to probe the conformational dynamics and free energy profiles of unassembled tubulin dimers. They find that both GTP- and GDP-tubulin have an ensemble average of intermediate bending conformation, and that GTP-tubulin has much larger bending flexibility compared to GDP-tubulin. Igaev and Grubmuller conclude that increased bending flexibility and reduced free energy of kinking make GTP-tubulin more favorable for incorporation into the MT lattice. Finally, the authors also use functional mode analysis to identify two distinct bending modes for tubulin – twist-bending (TB) and splay-bending (SB) – are identified through functional mode analysis in both nucleotide states. Igaev and Grubmuller speculate that the transition from SB state to TB state when tubulin is bound at the assembling MT tip generates destabilizing strain that could contribute to the kinetics of the MT disassembly process. The ideas that nucleotide state controls tubulin flexibility, and that nucleotide-dependent differences in flexibility may be important for microtubule dynamics, are interesting and new. All reviewers felt that the work was technically performed at a very high level, and that a suitably revised version of the paper that satisfactorily addresses the concerns articulated below has the potential to make a nice contribution to the field.

Essential revisions:

1) While the work was very well presented, there was some concern that as written the manuscript might not be accessible to a broad audience. Additional efforts at simplification and clarification would be extremely helpful. 'Conformational mode' or 'reaction coordinate' axes are not always easy to relate to the known structures even given the black dots in Figure 3, Figure 4, and Figure 5 that show where the experimental structures fall. Figure 4—figure supplement 1 was a good example of helpful simplification of what are otherwise complicated concepts explained with too much technical jargon.

2) The results from the simulations, upon which all the analysis rests, are hard to assess: whether convergence has been reached and/or whether different equivalent simulations yield overlapping results, and how some of the conformations observed relate to the known structures is not easily discerned from what was presented. This should be addressable through additional panels (possibly supplemental) made from existing data: convergence or lack thereof could be illustrated using color coding or shading overlaid onto the pink and green distributions of Figure 3 and Figure 4 to show simulation time and/or to separate the equivalent simulations, and by annotating the rotation angle between a- and b-tubulin in Figure 3, Figure 4, and Figure 5, and possibly also by proving reference structures in the accompanying supplemental videos (which are helpful). Alternately, or in addition, it would be useful to show the extent of bending transition (conformational mode 1) along the simulation time as the evidence of the simulations are converged. Would the differences between Video 2 and Video 3 be better perceived in a side-by-side presentation?

3) The manuscript is in part framed around the idea that it is testing competing models for how nucleotide affects tubulin conformation: an 'allosteric' model positing that GTP-binding causes tubulin to become straighter, and a 'lattice' model positing that tubulin binding to the GTP lattice is what causes tubulin to become straighter. Strictly speaking, because the authors did not perform simulations of tubulin assemblies, their data may not really have much to say about the lattice model. Furthermore, some of the simulations start from 'straight' conformations that were taken from recent cryo-EM structures of microtubules in different nucleotide states. It seems that these simulations do not converge onto the curved conformation of unpolymerized tubulin (and conversely, that simulations starting from curved tubulin do not reach straight conformations). It is interesting that there appears to be such a large difference in free energy between curved and straight conformations, but at the same time the lack of connection between the two sets of simulations also leaves room for doubt about whether something is missing (or being missed). To address these issues surrounding the 'straight' simulations, the authors are requested to more explicitly acknowledge (i) the potential limitations of starting simulations from straight conformations of tubulin 'plucked' from the lattice (i.e. removed from its natural context where the conformation is stabilized by longitudinal and lateral interactions in the lattice), and (ii) the potential limitations of not having simulated assemblies of tubulin. Along these lines, the reviewers recommend removing or at least substantially condensing the penultimate paragraph (and associated discussion in Materials and methods section). It seemed overly detailed for how speculative it was, given the limitations articulated here along with a lack of simulations of polymerization dynamics.

4) In Figure 7B, the uncertainty of the free energy profile is large at the two local minimums and nearly zero at the transition state around 2.5 nm. This seems counterintuitive. The authors should also evaluate the uncertainty using the block average method and compare with the results presented here. In Figure 8, the transition from SB state to TB state when GTP dimer has arrived at the MT tip requires an activation energy of around 20 kT, assuming the free energy profile in Figure 7B is correct. This seems to be much too high considering the MT assembly speed to be around 2 μm/min, or 50 tubulin dimers/sec. Could the authors justify the model using existing experimental data?

---

## [Author Response]

Essential revisions:1) While the work was very well presented, there was some concern that as written the manuscript might not be accessible to a broad audience. Additional efforts at simplification and clarification would be extremely helpful. 'Conformational mode' or 'reaction coordinate' axes are not always easy to relate to the known structures even given the black dots in Figure 3, Figure 4, and Figure 5 that show where the experimental structures fall. Figure 4—figure supplement 1 was a good example of helpful simplification of what are otherwise complicated concepts explained with too much technical jargon.

We thank the reviewer for pointing out that not enough attention was paid to explanation and simplification of complex mathematical aspects and to establishing the link between known structures and reaction coordinate values. We have now rewritten and extended the middle part of the first Results section as well as Figure 2. More specifically:

Figure 2 has been extended to include an auxiliary panel (A) showing a “topological map” of known tubulin structures that were involved in our study as well as structures that are widely used in literature to demonstrate different curvature states of tubulin.Three supplementary videos accompanying Figure 2 have been provided to visualize the full set of experimental structures (Figure 2—supplementary video 1) and the structural motions along conformational modes 1 and 2 (Figure 2Figure 2—supplementary video 2 and Figure 2—supplementary video 3).To further improve the visual link between experimental structures and simulations, we have additionally indicated the positions of the PDB structures used in the free MD simulations in all subsequent figures (Figures 3–7).Following the example of Figure 4—figure supplement 1 (simplified explanation of the ensemble separation search), a similar supplementary figure (Figure 2—figure supplement 1) has been added that explains the main idea behind the PCA using a 2D example.

2) The results from the simulations, upon which all the analysis rests, are hard to assess: whether convergence has been reached and/or whether different equivalent simulations yield overlapping results, and how some of the conformations observed relate to the known structures is not easily discerned from what was presented. This should be addressable through additional panels (possibly supplemental) made from existing data: convergence or lack thereof could be illustrated using color coding or shading overlaid onto the pink and green distributions of Figure 3 and Figure 4 to show simulation time and/or to separate the equivalent simulations, and by annotating the rotation angle between a- and b-tubulin in Figure 3, Figure 4, and Figure 5, and possibly also by proving reference structures in the accompanying supplemental videos (which are helpful). Alternately, or in addition, it would be useful to show the extent of bending transition (conformational mode 1) along the simulation time as the evidence of the simulations are converged. Would the differences between Video 2 and Video 3 be better perceived in a side-by-side presentation?

The reviewer requests to include the information on convergence of the simulated ensembles and the calculated free energy profiles. We have now assessed the convergence and extended the corresponding Figure 3, Figure 6, and Figure 7. Our analysis confirms that the independent equivalent simulations for every combination of starting conformation and nucleotide state yield consistently overlapping results. As Figure 4 and Figure 5B rely on the same simulated ensembles as Figure 3 (different representation of the same data), convergence for these plots needs no assessment. Convergence assessment for the free energy profiles shown in Figure 6 and Figure 7B is now provided as supplementary figures (see also (4) (I) for more details). The second point – how some of the conformations observed compare with known structures – closely relates to (1) and has been addressed as described above.

We also thank the reviewer for the very helpful suggestions on how to improve this part of the manuscript and have now introduced the following modifications:

Figure 3—figure supplement 1 has been added that shows the level of convergence by overlaying point clouds originating from different independent simulations. In the case of straight tubulin, 2 independent simulations (~3 µs each) were carried using the same starting structures (3JAT for GTP-tubulin, 3JAS for GDP-tubulin). In the case of curved tubulin, 2 independent simulations (~3 µs each) were carried out using 4 starting structures (5JQG/5FNV for GTP-tubulin, 4ZOL/4ZHQ for GDP-tubulin).Figure 6—figure supplement 1 and Figure 7—figure supplement 1 have been added that show individual overlapping umbrella sampling histograms and the uncertainty of the free energy profiles assessed using an independent error estimation method.Videos showing the TB and SB bending motions have been merged to create a side-by-side representation, as suggested by the reviewer (Figure 5—video supplement 1).

3) I) The manuscript is in part framed around the idea that it is testing competing models for how nucleotide affects tubulin conformation: an 'allosteric' model positing that GTP-binding causes tubulin to become straighter, and a 'lattice' model positing that tubulin binding to the GTP lattice is what causes tubulin to become straighter. Strictly speaking, because the authors did not perform simulations of tubulin assemblies, their data may not really have much to say about the lattice model.

The reviewer points out that, because no simulations of tubulin assemblies were performed, our simulations may not say much about the lattice model. While it is true that we only performed simulations of individual tubulin dimers in solution, we respectfully disagree with the claim that these would not allow conclusions on the lattice model.

Generally, the logic is as follows: for the canonical lattice or allosteric model to hold, certain properties of unassembled dimers are required – those have been tested in our work. More specifically, the canonical lattice model (Andreu et al., 1989; Buey et al., 2006; Rice et al., 2008) postulates that (a) soluble tubulin is kinked irrespectively of the nucleotide state, and (b) lateral and longitudinal bond formation in the MT lattice provides the necessary free energy to straighten incoming dimers. On the contrary, the canonical allosteric model (Melki et al., 1989; Müller-Reichert et al., 1998; Wang and Nogales, 2005) postulates that (a) soluble GDP-tubulin is kinked in solution, and (b) it is GTP-binding, not the lattice, what triggers a kinked-to-straight transition, which makes GTP-dimers more compatible with MT geometry. Therefore, an observed straightening of GTP-tubulin in solution compared to GDP-tubulin, or any other allosteric response, would indeed rule out the lattice model, at least in its original form. Assessing the dynamics and energetics of both GTP- and GDP-tubulin in solution, hence, directly and simultaneously tests both assembly models. Because only little is known up to now about the conformational flexibility of free tubulin, both models are still possible, which motivated us to focus on single tubulin dimers.

We think the reviewer got misled by the paragraph 4 of the Introduction (related to Figure 1), where the above reasoning and the critical point leading to a divergence of the canonical models may have been not sufficiently articulated. We have now filled this logical gap in the Introduction by more explicitly linking the problem (contradictory evidence for both models) to the proposed solution (assessing the dynamics and energetics of unassembled GTP- and GDP-tubulin on a multiple-µs timescale).

II) Furthermore, some of the simulations start from 'straight' conformations that were taken from recent cryo-EM structures of microtubules in different nucleotide states. It seems that these simulations do not converge onto the curved conformation of unpolymerized tubulin (and conversely, that simulations starting from curved tubulin do not reach straight conformations). It is interesting that there appears to be such a large difference in free energy between curved and straight conformations, but at the same time the lack of connection between the two sets of simulations also leaves room for doubt about whether something is missing (or being missed).

The reviewer is puzzled by the fact that the two sets of simulations in Figure 3 (starting from straight and kinked conformations) do not converge onto each other – and so were we initially. Obviously, for long enough simulations, the two should converge. Therefore, one would expect a free energy barrier between the two regions of conformational space that is too high to be overcome within our multiple microseconds simulations. Without this barrier, the green and pink ensembles in Figure 3 would quickly merge in an intermediate region, and so there must be a barrier preventing this merging within microseconds. As we also felt that there might be something missing, we proceeded to the more in-depth analysis shown in Figures 4–7. This included the derivation of reaction coordinates that simultaneously take into account both the known experimental structures and our simulated ensembles.

And indeed, the missing piece of information was a second type of conformational transition in the simulated ensembles described structurally and energetically in Figure 7, but unfortunately “hidden” in Figure 3 because conformational modes 1 and 2 were derived by a PCA using only a limited set of experimental structures. Therefore, Figure 4 gives a much more complete representation of the dimer dynamics as it: (a) relies on experiment and simulation, and (b) reflects both the largest-amplitude motion in the system (bending) and the most energetically expensive motion (the new hidden motion).

We assume it has not become sufficiently clear from our initial text that the actual bending motion in Figure 4 is characterized by the vertical axis and, in fact, well sampled in our simulations, whereas the new motion characterized by the horizontal axis is poorly sampled due to a high free energy barrier. We have therefore rewritten the last paragraph of the subsection (“Free tubulin dynamics depend on the nucleotide state”) as well as the subsection (“Trajectory analysis reveals two states with different bending modes”).

III) To address these issues surrounding the 'straight' simulations, the authors are requested to more explicitly acknowledge (i) the potential limitations of starting simulations from straight conformations of tubulin 'plucked' from the lattice (i.e. removed from its natural context where the conformation is stabilized by longitudinal and lateral interactions in the lattice), and (ii) the potential limitations of not having simulated assemblies of tubulin. Along these lines, the reviewers recommend removing or at least substantially condensing the penultimate paragraph (and associated discussion in Materials and methods section). It seemed overly detailed for how speculative it was, given the limitations articulated here along with a lack of simulations of polymerization dynamics.

(i) See the discussion in (II). (ii) See the discussion in (I).

Paragraph 4 of the Introduction has been modified to account for the discussion in point 3.We now provide an improved transition between subsection “(“Free tubulin dynamics depend on the nucleotide state” and subsection (“Trajectory analysis reveals two states with different bending modes” as well as a rewritten version subsection (“Trajectory analysis reveals two states with different bending modes”).To avoid confusion with respect to Figure 4, we have renamed the “optimal reaction coordinate (RC)” into the “ensemble separation RC”. This should underline its conceptual difference from the “bending RC” and still communicate its role – yielding best separation between two simulated ensembles in terms of overlap.The penultimate paragraph of the Discussion section and the associated paragraph in Materials and methods section have been removed, as suggested by the reviewers.

4) In Figure 7B, the uncertainty of the free energy profile is large at the two local minimums and nearly zero at the transition state around 2.5 nm. This seems counterintuitive. The authors should also evaluate the uncertainty using the block average method and compare with the results presented here.

We agree with the reviewer that this representation of free energy uncertainties is a bit confusing, albeit not wrong *per se*. Since any free energy is defined up to an arbitrary additive constant, only free energy differences matter. Hence, the only relevant uncertainty is that of a free energy difference rather than that of an absolute free energy value. As a consequence, one needs to set a “reference point” along the reaction coordinate at which the absolute free energy value and the corresponding uncertainty are both zero. The error bars at all other point of the free energy profile will then indicate the errors of free energy differences relative to this reference point. In the case at hand (Figure 7B), we are much more interested in the uncertainty of the barrier height with respect to the free energy minima. It is indeed much better to set the reference point in one of the minima, which we have now done. We think this clarifies the issue.

As also requested by the reviewer, we have now evaluated the uncertainty of the free energy profile in Figure 7B using a conceptually unrelated method.

In Figure 8, the transition from SB state to TB state when GTP dimer has arrived at the MT tip requires an activation energy of around 20 kT, assuming the free energy profile in Figure 7B is correct. This seems to be much too high considering the MT assembly speed to be around 2 μm/min, or 50 tubulin dimers/sec. Could the authors justify the model using existing experimental data?

The reviewer touches upon one of the main messages of our study, namely: what is the meaning of the high energy barrier between the TB and SB states (Figure 7) and how does it agree with MT assembly? As we mentioned in the Discussion section, we cannot rigorously prove at the present stage that this new transition actually has a functional role, but we would be very much surprised if it did not. What is however possible is to make conclusions on the “compatibility” of this transition with MT assembly/disassembly based on our results and existing kinetic data.

We first note that this very high barrier between TB and SB seems not to disturb the assembly process – otherwise, MT would never polymerize. Indeed, dimers locked in MT bodies (3JAT/3JAS and others) are clearly attributed to the TB state and stay there for multiple µs in our simulations, while dimers crystallized under non-polymerizing conditions are clearly in the SB state and stay there for multiple µs as well (Figure 4). From this we conclude that spontaneous transition of free dimers in solution from SB to TB (Figure 7B) is very unlikely to be required for initial polymerization – it would require an activation energy of ~20 kT and imply a characteristic transition time of several seconds. This would clearly hinder the assembly process, given association times of dimers with the MT tip of ~0.7-1.4 ms at 12-24 µM free tubulin or, equivalently, MT assembly speeds of 15-30 nm/s (Gardner et al., 2011, both studies).

Therefore, we think there are two possible scenarios of how this barrier is overcome during polymerization. The first possibility is sketched in Figure 8 (top cycle) and assumes that the activation energy to cross the SB-TB barrier for a newly arrived dimer at the MT tip comes from the lattice neighbors, particularly, from favorable lateral and longitudinal bond formation. In other words, “peer pressure” would override the tendency of each dimer to be kinked and stay in the SB state (Figure 6 and Figure 7). The second possibility is that both SB and TB dimers are equally well incorporated into the MT lattice. This transition may be accomplished much later in the MT body after the dimer incorporation has completed and, hence, become relevant only when the MT starts to disassemble.

While we cannot discriminate between the two possibilities at present, we favor the first for two reasons. First, no tubulin structure derived from the MT body, for example, far away from the dynamic tip, has (yet) been deposited showing a dimer in the SB state. Second, the values for longitudinal and lateral bond energies found in literature (VanBuren et al., 2002, VanBuren et al., 2005, Gardner et al., 2011, Castle and Odde, 2013, Kononova et al., 2014) seem to be sufficient to antagonize the high SB-TB barrier of ~20 kT. In particular, the longitudinal bond energy ranges between -16.8 kT and -25.2 kT (not including the entropic cost of binding from the bulk to the MT tip which is ~10 kT (Erickson and Pantaloni, 1981; Castle and Odde, 2013)), while the lateral bond energy is estimated to be -3.6 – -11.7 kT. Taking the lower bounds of these estimates, one arrives at a combined (one lateral and one longitudinal) energy of -20.4 kT, which is already enough to counterbalance the high SB-TB activation energy.

Thus, the presence of the barrier is most probably compatible with MT assembly, and this transition is most probably enforced by the lattice neighbors during initial polymerization (lattice binding and subsequent interdimer bond formation).

The reference point in Figure 7B is now placed at the minimum corresponding to the TB state (green).A supplementary figure (Figure 7—figure supplement 1A) has been added that shows overlapping umbrella sampling histograms along the ensemble separation RC.We have also evaluated the uncertainty of the profile in Figure 7B using a conceptually different method proposed by Zhu and Hummer (2012) which relies on the statistical error of the mean force in each individual umbrella window. These errors, in turn, are obtained from block averages. Our calculations yielded values that are largely consistent with the error estimates obtained through Bayesian bootstrapping (Figure 7—figure supplement 1B). Details on this type of error estimation have been added to subsection “Free energy calculations” in Materials and methods.We have added a paragraph in the Discussion section to include the current limitation of not knowing the assembly stage at which the dimer transits from SB to TB state, as well as a short discussions of the above two scenarios.